# Improved ethanol electrooxidation performance by shortening Pd–Ni active site distance in Pd–Ni–P nanocatalysts

Lin Chen[1], Lilin Lu[2], Hengli Zhu[1], Yueguang Chen[1,3], Yu Huang[4], Yadong Li[3] & Leyu Wang[1]

Incorporating oxophilic metals into noble metal-based catalysts represents an emerging strategy to improve the catalytic performance of electrocatalysts in fuel cells. However, effects of the distance between the noble metal and oxophilic metal active sites on the catalytic performance have rarely been investigated. Herein, we report on ultrasmall ($\sim 5$ nm) Pd–Ni–P ternary nanoparticles for ethanol electrooxidation. The activity is improved up to 4.95 A per $mg_{Pd}$, which is 6.88 times higher than commercial Pd/C (0.72 A per $mg_{Pd}$), by shortening the distance between Pd and Ni active sites, achieved through shape transformation from Pd/Ni–P heterodimers into Pd–Ni–P nanoparticles and tuning the Ni/Pd atomic ratio to 1:1. Density functional theory calculations reveal that the improved activity and stability stems from the promoted production of free OH radicals (on Ni active sites) which facilitate the oxidative removal of carbonaceous poison and combination with $CH_3CO$ radicals on adjacent Pd active sites.

[1] State Key Laboratory of Chemical Resource Engineering, Beijing University of Chemical Technology, Beijing 100029, China. [2] School of Chemistry and Chemical Engineering, Wuhan University of Science and Technology, Wuhan 430081, China. [3] Department of Chemistry, Tsinghua University, Beijing 100086, China. [4] Department of Materials Science & Engineering, University of California Los Angeles, California 90095, USA. Correspondence and requests for materials should be addressed to L.Y.W. (email: lywang@mail.buct.edu.cn).

Direct fuel cells have been recognized as a promising future power source due to advantages, including environmental aspects, facile storage, easy refilling and high power density[1–7]. However, the lack of active and durable anode catalysts has greatly limited the large-scale commercialization of direct fuel cells. So far, platinum (Pt) has been considered as one of the best catalysts and exclusively utilized in fuel cells[2–6], but they suffer from high cost and poor carbon monoxide (CO) tolerance[8–10]. Alloying Pt with less expensive oxophilic metals ($M$) such as gold (Au), silver (Ag) and especially nonprecious 3d transition metals[11–14] is an effective route to improve CO tolerance and catalytic activity of catalysts, owing to the synergistic and electronic structure alteration mechanism[15–24]. However, the dissolution of these metals remains the major reason for the severe degradation of the catalytic performance of these alloyed catalysts.

Compared with Pt-based catalysts, palladium (Pd) is more efficient for ethanol oxidation reaction (EOR) in alkaline media due to its relatively high catalytic activity, lower cost and better resistance to CO poisoning[7,25]. By fabricating Pd–$M$–P ternary phosphide nanoparticles (NPs)[26–28], the stability and activity of the catalysts can be remarkably improved. The introduction of oxophilic metals such as Ni (Ru, Rh, Sn or Ag) facilitates the formation of OH radicals and drives the EOR without the production of poisoning by-products such as CO[4,26]. These OH radicals formed on Ni sites then combine $CH_3CO$ radicals on the adjacent Pd active sites to generate acetate ions, and this combination has been confirmed as the rate-determining step for EOR[10,13]. Therefore, simultaneously increasing the Ni and Pd active sites and shortening the distance between these two active sites in multicomponent catalysts holds promising potential to permit the absorption and desorption on the surface and acceleration of mass transfer between different active sites during catalytic progress. Despite tremendous studies have been made to downsize the noble metals to an atomically dispersed (single-atom) catalyst to maximize the activity of noble metals[1,15,20,29–31], efforts have rarely been paid to simultaneously increase the noble metal (Pd) and oxophilic metal (Ni) active sites. Thus, the controlled synthesis of ultrasmall Pd–Ni–P ternary NPs with rich and adjacent Pd and Ni active sites is still challenging but highly attractive to increase the EOR catalytic performance.

Herein we report a two-step solvothermal strategy for the synthesis of small ($\sim 5\,nm$) Pd–Ni–P ternary NPs with tunable Ni/Pd atomic ratio and controlled distance between Pd and Ni active sites (termed as Pd–Ni distance). Briefly, the NPs are synthesized via thermolysis ($260\,°C$, $1\,h$) of the mixture containing $Pd(acac)_2$, $Ni(acac)_2$ and trioctylphosphine, and then treated at $290\,°C$ for $1\,h$. The NPs with a Ni/Pd atomic ratio of 1:1 show the best catalytic performance towards EOR. By tuning the phosphorization temperature and time, the NPs are transformed from Pd/Ni–P heterodimers into Pd–Ni–P NPs with closer Pd–Ni distance (Fig. 1). Besides dramatically improved durability, the EOR activity is substantially enhanced from 4.12 to 4.42 A per $mg_{Pd}$ and finally 4.95 A per $mg_{Pd}$, which are 5.72, 6.14 and 6.88 times compared with that of commercial Pd/C (0.72 A per $mg_{Pd}$), respectively.

## Results

**Characterizations of Pd–Ni–P nanocatalysts.** As shown in the transmission electron microscopy (TEM) images (Fig. 2a–c), all the NPs are well dispersed with sizes of $\sim 5\,nm$ (the size distribution is shown in Supplementary Fig. 1) despite a slight decrease after prolonging the phosphorization time from Fig. 2a ($Pd_{38}Ni_{49}P_{13}$, $5.5 \pm 1.0\,nm$) to Fig. 2b ($Pd_{38}Ni_{45}P_{17}$, $5.3 \pm 1.0\,nm$) and Fig. 2c ($Pd_{40}Ni_{43}P_{17}$, $5.3 \pm 0.5\,nm$). Different from the reported phosphides using $NaH_2PO_2$ and $NaBH_4/N_2H_4$ as precursors[26,27,32,33], these as-prepared NPs are very small with rich and adjacent Ni and Pd active sites. From the high-resolution TEM (HRTEM) image, it is clear that the NPs are Pd/Ni–P heterodimers when the phosphorization (at $260\,°C$) time is only 5 min (Fig. 2d and Supplementary Fig. 2a). By prolonging the phosphorization (at $260\,°C$) time to 1 h, the heterodimers were alloyed into $Pd_{38}Ni_{45}P_{17}$ NPs. It is noteworthy that if the NPs were treated with one-step strategy ($290\,°C$, $2\,h$), the particle size would increase and the size distribution would become wider. Therefore, we used the two-step strategy to fabricate the NPs with different Pd–Ni distance (alloying degree). According to the clear lattice fringes shown in the HRTEM image, the Pd species still existed as large domains in the Ni–P matrixes (Fig. 2e and Supplementary Fig. 2b). If the $Pd_{38}Ni_{45}P_{17}$ NPs were phosphorized for another 1 h at $290\,°C$, the Pd domains became smaller and the Pd–Ni distance was further decreased (Fig. 2f and Supplementary Fig. 2c). Correspondingly, the clear lattice fringes became invisible, suggesting that these $Pd_{40}Ni_{43}P_{17}$ NPs were amorphous, which was in accord with other reports[28]. This amorphous structure was also confirmed by X-ray diffraction (XRD) analysis. As summarized in Supplementary Fig. 3a, most of Pd elements exist as Pd(0) species with minor $Pd_2Ni_2P$ species, which was further confirmed by the X-ray photoelectron spectroscopy (XPS) analysis detailed later. In comparison, the NPs with only 0.4% of Pd doping show very good crystallinity (Supplementary Fig. 3b), which can be indexed to $Ni_{12}P_5$ (JCPDS: 22-1190). Therefore, the amorphous structure of our NPs can be attributed to the Pd-doping that causes the lattice distortion.

The spatial distribution of Pd, Ni and P species in the Pd–Ni–P ternary NPs was further confirmed via high-angle annular dark-field image and elemental mapping (Fig. 2g–i and Supplementary Figs 4–6). As shown in Fig. 2g,h, there is distinct phase segregation in the composite maps of Pd and Ni. However, when the phosphorization time is further prolonged, no obvious phase segregation in the composite maps of Pd and Ni can be observed (Fig. 2i), which is in agreement with the result of HRTEM, indicating that the Pd is highly dispersed into the Ni–P matrix. From the elemental maps (Supplementary Figs 4–6), it also can be observed that the Ni and P species are homogeneously distributed

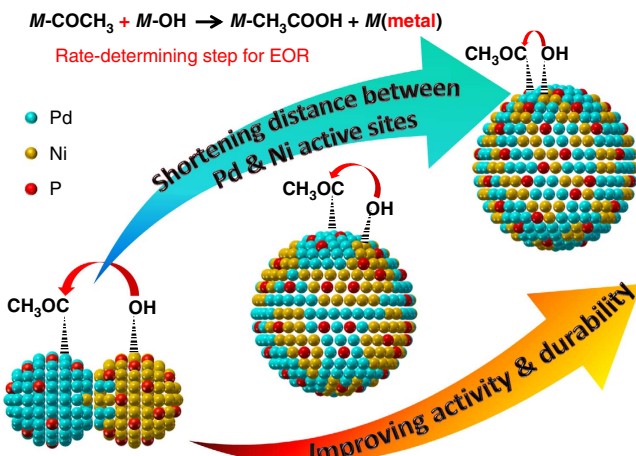

$M\text{-}COCH_3 + M\text{-}OH \longrightarrow M\text{-}CH_3COOH + M(\text{metal})$

Rate-determining step for EOR

**Figure 1 | Scheme for improving catalytic performance by shortening Pd–Ni active site distance.** Scheme of the reaction between $CH_3CO$ radical on Pd and OH radical on Ni, where the distance between Pd and Ni active sites is shortened from left to right by prolonging the phosphorization time of Pd–Ni–P nanocatalysts.

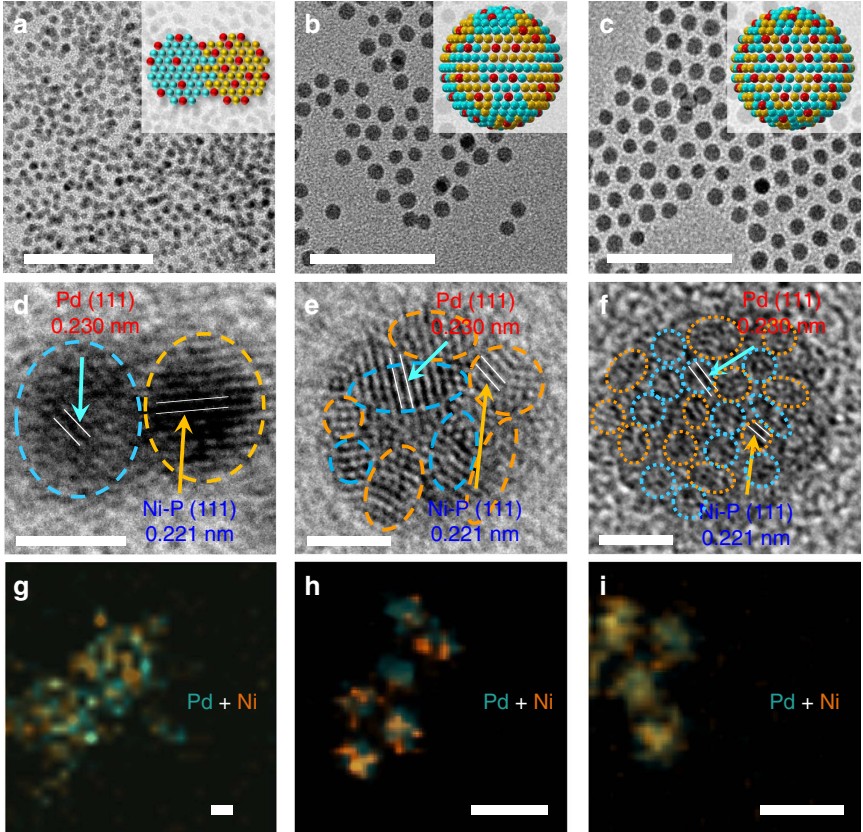

**Figure 2 | TEM and elemental mapping analysis.** TEM (**a–c**), HRTEM (**d–f**) images and elemental mapping (**g–i**) of Pd–Ni–P nanocatalysts with shortened distance between Pd and Ni active sites via increasing phosphorization time from 5 min (**a,d,g**, $Pd_{38}Ni_{49}P_{13}$, 260 °C) to 1 h (**b,e,h**, $Pd_{38}Ni_{45}P_{17}$, 260 °C) and then 2 h (**c,f,i**, $Pd_{40}Ni_{43}P_{17}$, 260 °C 1 h and 290 °C 1h). The atomic ratio of Ni/Pd in the metal precursor for nanocatalyst synthesis is 1:1. Detailed elemental maps and analysis are shown in Supplementary Materials. The chemical composition of the NPs is detected with the inductively coupled plasma mass spectrometry (ICP-MS). Scale bars in **a–c** (50 nm), in **d–f** (2 nm) and in **g–i** (10 nm).

and the Pd is dominantly presented as Pd (0) rather than Pd–P, which is in line with the XPS results. It is noted that the Ni content in the Pd–Ni–P ternary NPs is slightly decreased after further phosphorization (Supplementary Table 1), which may be attributed to the dissolution of Ni.

**Electrocatalytic performance tests**. The catalytic performance of the as-prepared Pd–Ni–P NPs was investigated and the results were compared with those of commercial Pd/C catalysts. As an important parameter for the assessment of active sites of catalysts, the electrochemically active surface area (ECSA) was first evaluated. The ECSA of these NPs can be calculated from the charge required for oxygen desorption, that is, from the area of the reduction peak of PdO in the cyclic voltammograms (CVs) in 1.0 M of NaOH[13]. Figure 3a shows CVs of $Pd_{38}Ni_{49}P_{13}$, $Pd_{38}Ni_{45}P_{17}$, $Pd_{40}Ni_{43}P_{17}$ and commercial Pd/C catalysts in the deaerated NaOH solution (1.0 M) at a scan rate of 100 mV s$^{-1}$. The ECSA (m$^2$ per g$_{Pd}$) values of these nanocatalysts are estimated according to the equation ECSA = $Q/(0.405 \times W_{Pd})$, where $Q$ and $W_{Pd}$ are the coulombic charge by integrating peak area of the reduction of PdO (mC) and Pd loading (mg cm$^{-2}$) on the electrode, respectively. Meanwhile, 0.405 represents the charge required for the reduction of PdO monolayer (mC per cm$^2_{Pd}$). Herein, the ECSA values of these $Pd_{38}Ni_{49}P_{13}$, $Pd_{38}Ni_{45}P_{17}$ and $Pd_{40}Ni_{43}P_{17}$ NPs are 56.74, 57.36 and 63.22 m$^2$ per g$_{Pd}$, respectively. Interestingly, the $Pd_{40}Ni_{43}P_{17}$ NPs demonstrate the largest ECSA, which is 1.33 times higher than that of commercial Pd/C (47.50 m$^2$ per g$_{Pd}$). As expected,

the ECSA, *i.e.*, the activity significantly increased with the decrement of the distance between Pd and Ni active sites.

In addition, as shown in Fig. 3b, the catalytic activity of $Pd_{40}Ni_{43}P_{17}$ NPs initially enhanced with the increase of cycle numbers as previous report of amorphous electrocatalyst[14], and the maximum and stable peak current density appeared at the seventh cycle. Then all the nanocatalysts were activated by CV scanning for seven cycles before electrocatalysis evaluation. Electrocatalytic performance of these ternary NPs and commercial Pd/C catalysts for EOR was investigated in the solution of 1.0 M NaOH and 1.0 M $C_2H_5OH$ at 100 mV s$^{-1}$. Figure 3c shows the representative CVs for the EOR with different catalysts, in which the characteristic ethanol oxidation peaks are identified in the forward and backward scans. The mass peak current densities (normalized to the mass of Pd) of $Pd_{38}Ni_{49}P_{13}$ (4.12 A per mg$_{Pd}$), $Pd_{38}Ni_{45}P_{17}$ (4.42 A per mg$_{Pd}$) and $Pd_{40}Ni_{43}P_{17}$ (4.95 A per mg$_{Pd}$) NPs are almost 5.72, 6.14, and 6.88 times higher than that of commercial Pd/C (0.72 A per mg$_{Pd}$). Clearly, the $Pd_{40}Ni_{43}P_{17}$ NPs have the highest mass catalytic activity among these three nanocatalysts at a certain potential (0.8 V). Furthermore, the $Pd_{40}Ni_{43}P_{17}$ NPs show much slower current decay over time than $Pd_{38}Ni_{49}P_{13}$, $Pd_{38}Ni_{45}P_{17}$ NPs and commercial Pd/C (Fig. 3d), demonstrating an excellent stability. After 2,000 s chronoamperometry measurements, the mass activity of $Pd_{40}Ni_{43}P_{17}$ NPs (215.4 mA per mg$_{Pd}$) still maintain 5.36 times of commercial Pd/C (40.21 mA per mg$_{Pd}$) electrocatalysts. Moreover, further extending to 20 h, still $Pd_{40}Ni_{43}P_{17}$ NPs displayed better mass activity than that of Pd/C (Supplementary Fig. 7). All the results suggest that the

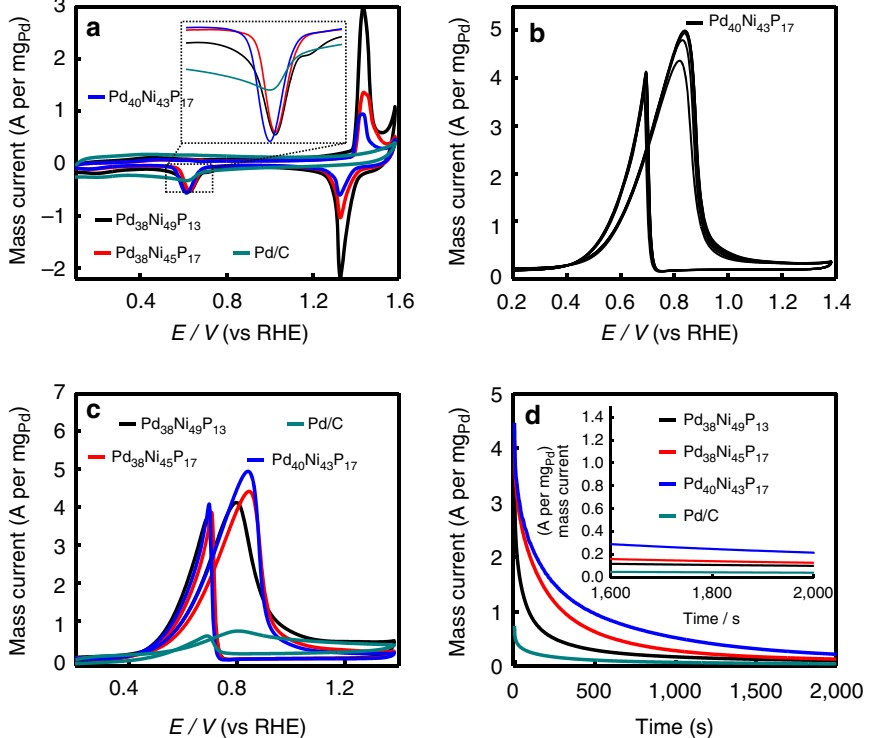

**Figure 3 | Catalytic activity and durability evaluation.** (**a**) CVs obtained on Pd–Ni–P NPs with different Pd–Ni distance, and commercial Pd/C in 1.0 M NaOH solution at a scan rate of 100 mV s$^{-1}$; (**b**) CVs of $Pd_{40}Ni_{43}P_{17}$ from first to seventh cycle in solution of 1.0 M NaOH and 1.0 M $C_2H_5OH$ at 100 mV s$^{-1}$; (**c**) CVs of Pd–Ni–P NPs with different Pd–Ni distance, and commercial Pd/C in 1.0 M NaOH and 1.0 M $C_2H_5OH$ at 100 mV s$^{-1}$; (**d**) chronoamperometry curves measured in 1.0 M NaOH and 1.0 M $C_2H_5OH$ (the corresponding potential was held at 0.8 V during the measurements).

$Pd_{40}Ni_{43}P_{17}$ NPs are comparable to or even more active and stable than many Pd-based EOR catalysts (Supplementary Table 2). It is noteworthy that further prolonging the phosphorization time will cause more Ni dissolution (Supplementary Table 1) and thus activity loss of nanocatalysts (Supplementary Fig. 8), suggesting that the Ni/Pd atomic ratio around 1:1 is favourable for a good catalytic activity and long durability. In another aspect, as comparison, the Ni–P and Pd–Ni NPs were also prepared and characterized and their catalytic activity toward EOR was also evaluated. As expected, the as-prepared $Ni_{12}P_5$ NPs did not show any catalytic activity toward EOR since the Pd domains are the active sites for EOR (Supplementary Fig. 9). With respect to the preparation of Pd–Ni alloy NPs, it was found that merely Pd–Ni aggregates (Supplementary Fig. 10) were obtained without utilization of TOP, not to mention its performance toward EOR.

**Effects of Ni/Pd atomic ratio on catalytic performance.** We further evaluated the influence of Ni/Pd ratio in the metal precursors on the catalytic performance (nanocatalyst synthesis conditions were shown in Supplementary Tables 3 and 4). With the increase of Ni/Pd ratio in metal salt precursors from 4/6 (Fig. 4a,d) to 5/5 (1:1) (Fig. 4b,e) and further 6/4 (Fig. 4c,f), the NPs became more and more uniform with slight increase in size. According to the ICP-MS test results, the compositions of these as-prepared products are $Pd_{47}Ni_{36}P_{17}$ (Fig. 4a, 4.5 ± 0.8 nm), $Pd_{38}Ni_{45}P_{17}$ (Fig. 4b, 5.3 ± 1.0 nm) and $Pd_{31}Ni_{53}P_{16}$ (Fig. 4c, 6.3 ± 1.5 nm), respectively. After further phosphorization, the compositions of these NPs are changed into $Pd_{54}Ni_{30}P_{16}$ (Fig. 4d, 3.9 ± 0.5 nm), $Pd_{40}Ni_{43}P_{17}$ (Fig. 4e, 5.3 ± 0.5 nm) and $Pd_{32}Ni_{50}P_{18}$ (Fig. 4f, 5.6 ± 1.0 nm), respectively. Significantly, the Ni content in the Pd–Ni–P ternary NPs decreases after further phosphorization,

which may be attributed to the dissolution of Ni. Meanwhile, the P content has no obvious change.

The catalytic performance of the as-prepared Pd–Ni–P NPs was investigated and the results were compared with those of commercial Pd/C catalysts. As shown in Fig. 5a, the ECSA values of these $Pd_{47}Ni_{36}P_{17}$, $Pd_{38}Ni_{45}P_{17}$ and $Pd_{31}Ni_{53}P_{16}$ NPs are calculated to be 44.38, 57.36 and 51.22 m$^2$ per $g_{Pd}$, respectively. Interestingly, the $Pd_{38}Ni_{45}P_{17}$ NPs demonstrate the largest ECSA, which is larger than that of commercial Pd/C catalysts (47.50 m$^2$ per $g_{Pd}$). After further phosphorization (290 °C, 1 h), the ECSA values of $Pd_{54}Ni_{30}P_{16}$, $Pd_{40}Ni_{43}P_{17}$ and $Pd_{32}Ni_{50}P_{18}$ NPs are 37.51 m$^2$ per $g_{Pd}$, 63.22 m$^2$ per $g_{Pd}$ and 51.76 m$^2$ per $g_{Pd}$, respectively. As shown in Fig. 5d, the $Pd_{40}Ni_{43}P_{17}$ NPs demonstrate the largest ECSA, which is 1.33 times higher than that of commercial Pd/C. All the results suggest that the Pd/Ni ratio around 5/5 in the salt precursor is preferable for good electrocatalytic performance.

Electrocatalytic performance of these ternary NPs and commercial Pd/C catalysts for EOR was investigated in the solution of 1.0 M NaOH and 1.0 M $C_2H_5OH$ at 100 mV s$^{-1}$. Figure 5b shows the representative CVs for the EOR with different catalysts. The mass peak current densities (normalized to the mass of Pd) of $Pd_{47}Ni_{36}P_{17}$ (4.09 A per $mg_{Pd}$), $Pd_{38}Ni_{45}P_{17}$ (4.42 A per $mg_{Pd}$) and $Pd_{31}Ni_{53}P_{16}$ (2.74 A per $mg_{Pd}$) NPs are almost 5.68, 6.1 and 3.81 times higher than that of commercial Pd/C (0.72 A per $mg_{Pd}$). Obviously, the $Pd_{38}Ni_{45}P_{17}$ NPs have the highest mass catalytic activity among the precursor NPs at a certain potential. The potential was held at 0.8 V during the measurements. Furthermore, the $Pd_{38}Ni_{45}P_{17}$ NPs show much slower current decay over time than $Pd_{47}Ni_{36}P_{17}$, $Pd_{31}Ni_{53}P_{16}$ NPs and commercial Pd/C (Fig. 5c), suggesting that the $Pd_{38}Ni_{45}P_{17}$ NPs have a much better durability for ethanol oxidation.

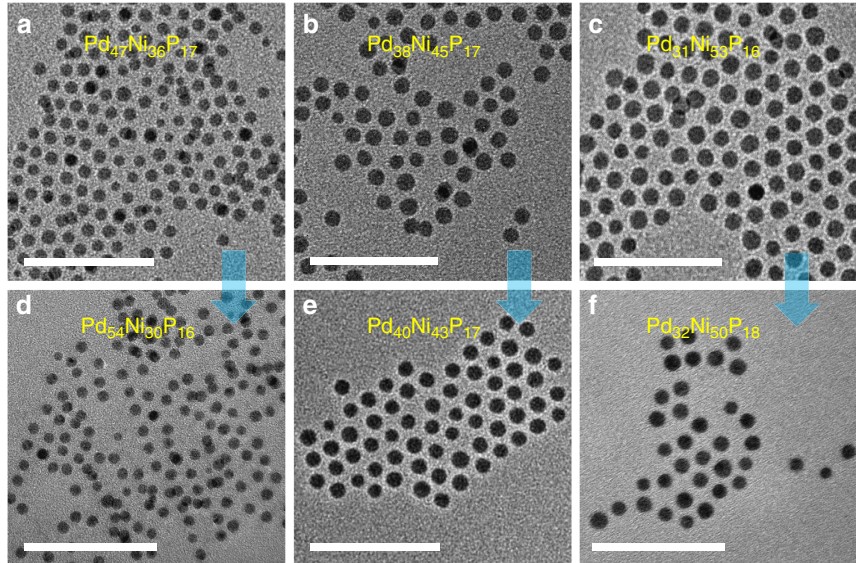

**Figure 4 | TEM images of ternary NPs with different Ni/Pd atomic ratios in the feeding precursors.** TEM images of Pd–Ni–P NPs (260 °C, 1 h) with different compositions (**a–c**). (**d,e,f**) are the resultant NPs of **a,b,c** with further phosphorization (290 °C, 1 h), respectively. The atomic ratio of Ni/Pd in the metal precursor is 4:6, 5:5 and 6:4 for (**a,d**), (**b,e**) and (**c,f**), respectively. Scale bars in **a–f**, 50 nm.

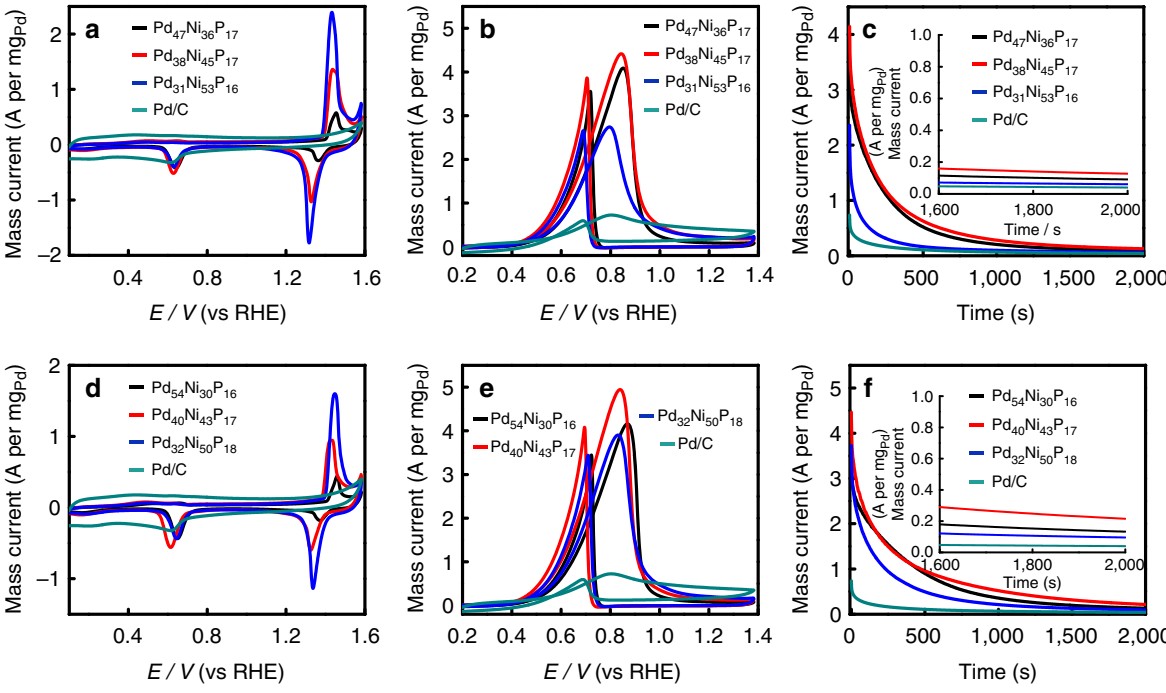

**Figure 5 | Catalytic performance evaluation of the NPs prepared with different Ni/Pd atomic ratios.** (**a,d**) CVs obtained on Pd–Ni–P NPs with different compositions and commercial Pd/C in 1.0 M NaOH solution at a scan rate of 100 mV s$^{-1}$; (**b,e**) CVs of Pd–Ni–P NPs with different compositions and commercial Pd/C in 1.0 M NaOH and 1.0 M $C_2H_5OH$ at 100 mV s$^{-1}$; (**c,f**) Chronoamperometry curves measured in 1.0 M NaOH and 1.0 M $C_2H_5OH$ (the corresponding potential was held at 0.8 V during the measurements).

It is found that the mass current density of the $Pd_{54}Ni_{30}P_{16}$ (4.16 A per $mg_{Pd}$), $Pd_{40}Ni_{43}P_{17}$ (4.95 A per $mg_{Pd}$) and $Pd_{32}Ni_{50}P_{18}$ (3.90 A per $mg_{Pd}$) NPs is always much higher than those of the corresponding $Pd_{47}Ni_{36}P_{17}$ (4.09 A per $mg_{Pd}$), $Pd_{38}Ni_{45}P_{17}$ (4.42 A per $mg_{Pd}$) and $Pd_{31}Ni_{53}P_{16}$ (2.74 A per $mg_{Pd}$) NPs, respectively (Fig. 5e). The best activity was obtained on $Pd_{40}Ni_{43}P_{17}$ (4.95 A per $mg_{Pd}$) NPs, which is 6.88 times higher than that of commercial Pd/C (0.72 A per $mg_{Pd}$), further demonstrating that the $Pd_{40}Ni_{43}P_{17}$ NPs own significantly enhanced electrocatalytic activity. Moreover, the $Pd_{40}Ni_{43}P_{17}$

NPs show a much better durability for ethanol oxidation than all other NPs and commercial Pd/C (Fig. 5f). The EOR test results indicate that the Ni/Pd ratio (in metal salt precursors) of 5/5 (1:1) shows the best electrocatalytic activity and durability. In addition, according to the TEM characterization, the nanocatalysts show no observable change in shape and size after electrocatalysis (Supplementary Fig. 11), accounting for the good stability. To further increase the Pd dispersion and investigate the catalytic performance, we then decreased the Pd/Ni ratio to 0.4% (ICP-MS result). But these NPs demonstrated a negligible catalytic activity

toward EOR (Supplementary Fig. 12), which may be attributed to the high sensitivity to CO poisoning of highly dispersed Pd[34]. It is noteworthy that because the Pd doping is too low, these NPs are indexed to $Ni_{12}P_5$ according to the XRD (Supplementary Fig. 3b) and HRTEM results (Supplementary Fig. 13).

**XPS analysis of the ternary nanocatalysts**. To investigate the mechanism of the satisfied electrochemical catalytic performance, the chemical states of Pd, Ni and P elements in both $Pd_{38}Ni_{45}P_{17}$ and $Pd_{40}Ni_{43}P_{17}$ NPs were accessed with XPS analysis (Fig. 6). High-resolution wide scans (Fig. 6a) indicated the presence of XPS peak for Pd, Ni and P. Deconvolution of the XPS spectra via peak fitting indicated that the $Pd3d_{5/2}$ (334–337 eV) and $3d_{3/2}$ (340–342 eV) peaks were observed (Fig. 6b). These peaks of Pd suggest that Pd(0) species predominates besides minor Pd(II) species on the nanocatalyst surface. The $3d_{5/2}$ peaks of Pd(0) in $Pd_{38}Ni_{45}P_{17}$ and $Pd_{40}Ni_{43}P_{17}$ NPs are located at 335.67 and 335.45 eV, respectively, and both of them positively shift as compared with pure Pd (0) (335.2 eV) reported previously[35]. Obviously, after further phosphorization, the $3d_{5/2}$ peak of Pd(0) in $Pd_{40}Ni_{43}P_{17}$ is negatively shifted 0.22 eV as compared with that in $Pd_{38}Ni_{45}P_{17}$ NPs (Fig. 6b). The positive shift of Pd $3d_{5/2}$ in binding energy suggests that the core-level of Pd shifts down with respect to the Fermi level of Pd, corresponding to a down-shift of the d-band centre of Pd due to the strong electron interactions involving Pd, Ni and P[26,36]. Furthermore, the broad Ni 2p peak is deconvoluted to five peaks that are assigned to three different oxidation states including $Ni(OH)_2$ (861.4 and 879.4 eV), NiO (855.8 and 873.4 eV) and Ni (852.5 and 869.6 eV) (Fig. 6c).

Obviously, in both $Pd_{38}Ni_{45}P_{17}$ and $Pd_{40}Ni_{43}P_{17}$ NPs, Ni is chiefly present in the form of NiO or $Ni(OH)_2$ apart from a few nominally reduced Ni species. Significantly, after further phosphorization, the Ni $2p_{3/2}$ (852.5 eV) in $Pd_{40}Ni_{43}P_{17}$ NPs is hardly observed, which can be attributed to the formation of Ni–P[28], and the dissolution of Ni confirmed by the ICP-MS results. For the P 2p spectrum shown in Fig. 6d, the peak at 133.1 and 129.7 eV of $Pd_{38}Ni_{45}P_{17}$ NPs could be assigned to the oxidized $P_2O_5$ and P(0) species, respectively, as reported in the literature[28,37]. The binding energy (BE) of P in $Pd_{38}Ni_{45}P_{17}$ and $Pd_{40}Ni_{43}P_{17}$ NPs shifts negatively by 0.7 and 0.8 eV, respectively, with respect to that of red phosphorus[37]. As shown in Fig. 6d, after further phosphorization, the binding energy of P(0) in $Pd_{40}Ni_{43}P_{17}$ NPs becomes more negative. These negative shifts in binding energy may be explained by assuming that P(0) species accepts partial electrons from surrounding Pd and Ni[26,38] which can be attributed to the high Pd dispersion and short Pd–Ni distance.

**CO anti-poisoning tests**. To further study the mechanism of good EOR performance, CO anti-poisoning experiments were also carried out. Chemisorbed CO intermediate has been identified as a major poison species for EOR on the active sites of catalysts[4]. CO stripping could serve as a model probe to evaluate the CO tolerance of catalysts[39,40]. It has been widely accepted that the CO stripping follows by the reaction between the formed $OH_{ads}$ and $CO_{ads}$ ($CO_{ads} + OH_{ads} \rightarrow CO_2 + H_2O$, $CO_{ads}$ and $OH_{ads}$ represent the radicals adsorbed on the active sites)[13,14]. Here, CO oxidation experiments were carried out at room

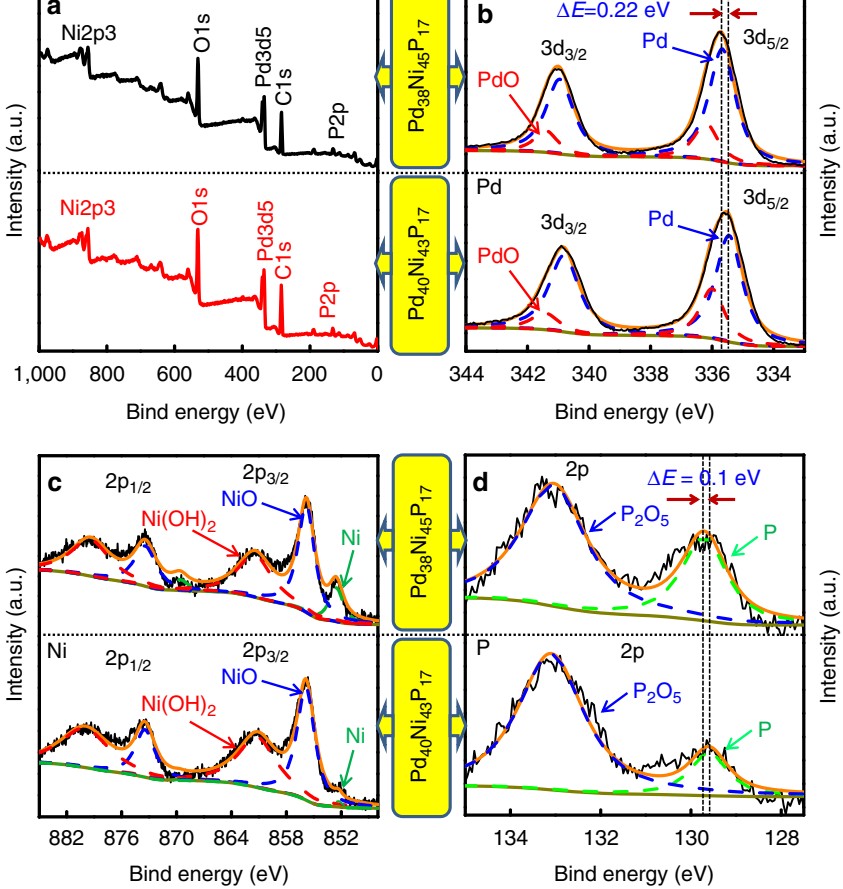

**Figure 6 | XPS analysis of the ternary nanoparticles.** XPS of $Pd_{38}Ni_{45}P_{17}$ and $Pd_{40}Ni_{43}P_{17}$ NPs (**a**); high-resolution region of Pd 3d (**b**), Ni 2p (**c**) and P 2p (**d**). All of the spectra were calibrated by C1s peak located at 284.8 eV.

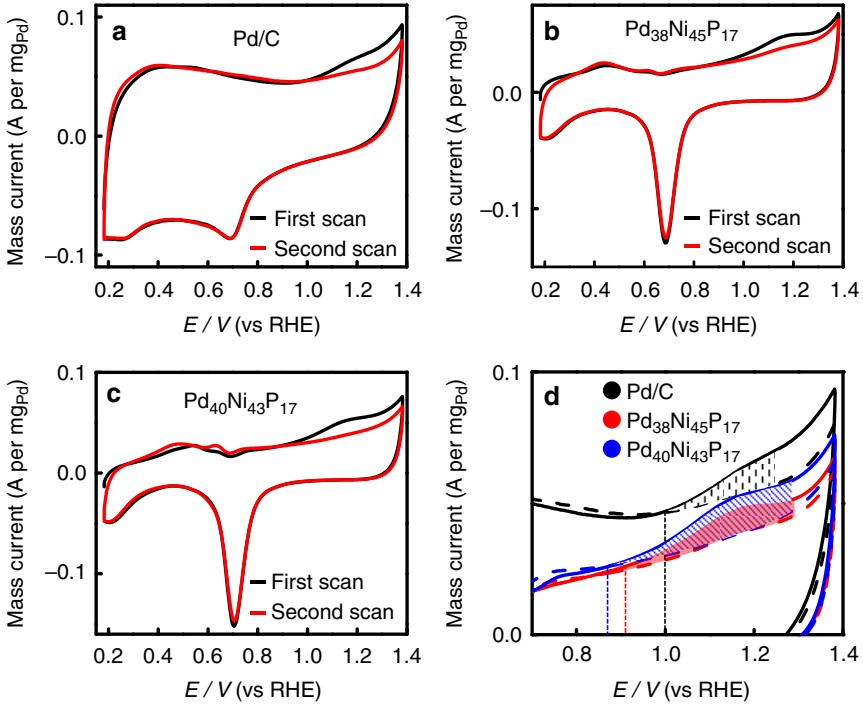

**Figure 7 | CO anti-poisoning tests.** CO stripping voltammograms for the Pd/C (**a**), $Pd_{38}Ni_{45}P_{17}$ (**b**) and $Pd_{40}Ni_{43}P_{17}$ NPs (**c**) in 1.0 M NaOH at a scan rate of 50 mV s$^{-1}$. In all cases, the red line is blank voltammetry, and the black line corresponds to a full CO coverage. (**d**) Overlap of CVs shown in **a,b,c** recorded within the potentials between 0.7 and 1.4 V.

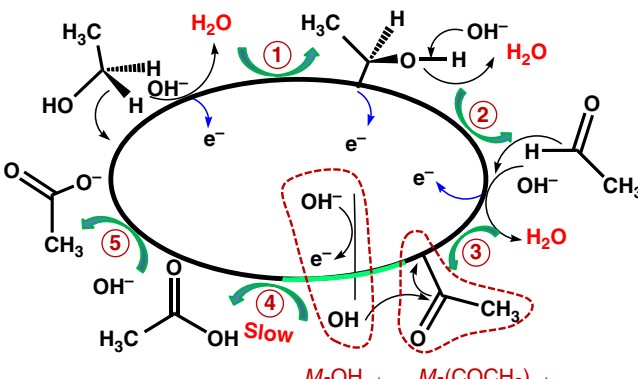

**Figure 8 | Scheme for the EOR process.** Proposed mechanism for the selective conversion of ethanol into acetate on Pd–Ni–P ternary nanocatalysts in alkaline media.

temperature in the NaOH solution (1.0 M) purged with nitrogen for 30 min and then bubbled with CO gas (99.9%) for 15 min. The potential was kept at 0.18 V to achieve the saturated coverage of CO at the Pd active sites. The residual CO in the solution was expelled by $N_2$ (99.99%) for 30 min. For simplicity, the weak CO anti-poisoning results of dumbbell $Pd_{38}Ni_{49}P_{13}$ NPs were not shown here. Figure 7 shows two consecutive CVs of Pd/C (Fig. 7a), $Pd_{38}Ni_{45}P_{17}$ (Fig. 7b) and $Pd_{40}Ni_{43}P_{17}$ NPs (Fig. 7c) recorded within the potentials between 0.18 and 1.38 V at a scan rate of 50 mV s$^{-1}$ in the saturated CO solution of 1.0 M NaOH. In the first forward scan, it is clear to see that a CO oxidation peak appears. On the second forward scan, the CO oxidation peak disappears owing to the complete elimination of CO on the surface of catalyst, indicating high CO anti-poisoning ability of these nanocatalysts. For easier comparison, Fig. 7d shows the overlap of these CVs recorded within the potentials between 0.7

and 1.4 V. It is clear that the onset and peak potentials for the electrocatalytic oxidation of CO on both $Pd_{40}Ni_{43}P_{17}$ and $Pd_{38}Ni_{45}P_{17}$ NPs are more negative than that on commercial Pd/C (1.0 V), indicating an increased CO oxidation activity and good CO anti-poisoning ability of these NPs[4,33,41]. In addition, the ECSA of Pd in the catalysts can be calculated from the area of the oxidation peak of $CO_{ads}$[42]. The active area of $Pd_{40}Ni_{43}P_{17}$ NPs is larger than that of $Pd_{38}Ni_{45}P_{17}$ and commercial Pd/C, which is in agreement with the electrocatalysis results of EOR. All the results further suggest that the $Pd_{40}Ni_{43}P_{17}$ NPs demonstrate excellent electrocatalytic performance for EOR and superior CO tolerance ability due to the shortened Pd–Ni distance.

**DFT calculations**. We further carried out the density functional theory (DFT) calculation (Supplementary Method 1) to understand the nature of the good electrocatalytic performance of the catalysts for EOR. It has been generally accepted that the EOR on metal (M) surfaces in alkaline media proceeds via the reactive-intermediate and/or the poisoning-intermediate (CO) pathway (Supplementary Fig. 14)[10,13]. In the reactive-intermediate pathway, the EOR on the nanocatalyst surface involves five steps and intermediates (Fig. 8), where the $M$-$OH_{ads}$ (*OH) and $M$-$(COCH_3)_{ads}$ (*CH_3CO) intermediates are combined to generate acetate anions. This combination between *OH and *CH_3CO has been confirmed as the rate-determining step for EOR[10,13]. In contrast, in poisoning-intermediate pathway, *CH_3CO solely decomposes into *CO and *CH_3[43], which block the active site and deteriorate the efficacy of the catalysts. Therefore, catalysts with facile *OH generation and high CO tolerance are in favour of excellent electrocatalytic performance towards EOR.

We constructed the cluster models of the nanocatalysts for DFT calculation to investigate the effects of incorporating Ni and P on the formation and dissociation of *OH on the catalysts. For the sake of space-saving, the details of discussion are shown

in Supplementary Materials. As shown in Supplementary Fig. 15, $OH^-$ shows preferential adsorption on Ni atoms, which is in good agreement with experimental observation[26]. This can be ascribed to the electrostatic attraction between $OH^-$ and Ni atom that carries significantly positive charge[26,28,44] demonstrated by Hirshfeld charge analysis (Supplementary Table 5). However, the P atom carries significant negative charge and Pd atom is almost electrically neutral in Pd–Ni–P ternary catalysts, which is in accord with our XPS results. The Pd–Ni–P ternary NPs show the highest adsorption energy for $OH^-$ ($115.7\,kcal\,mol^{-1}$) and the lowest desorption energy ($44.2\,kcal\,mol^{-1}$) for *OH, suggesting that the incorporation of Ni and P is favourable for both chemical absorption of $OH^-$ and desorption of free *OH, which thus facilitates the formation of $CH_3COOH$ (Fig. 8). Therefore, the incorporation of Ni and P in the ternary nanocatalysts drives the EOR preferentially through the efficient reactive-intermediate pathway.

## Discussion

In summary, we report a facile strategy to enhance the catalytic activity of small Pd–Ni–P ternary nanocatalysts by tuning the Ni/Pd atomic ratio to 1:1 and shortening the distance between Pd and Ni active sites. Our experimental and DFT calculation results highlight that the incorporation of Ni/P and the shortened distance between Pd and Ni active sites greatly facilitates the formation of free OH radicals and thus, speeds up the combination between OH and $CH_3CO$ radicals, that is, the rate-determining step for EOR. In addition, the CO anti-poisoning ability has also been enhanced, and therefore these ternary nanocatalysts achieve impressive EOR activity and long-term stability compared with commercial Pd/C catalysts. This research offer an interesting viewpoint to improve the catalytic activity and boost the durability by simultaneously increasing the noble metal and oxophilic metal active sites and shortening the distance between these two kinds of active sites in multicomponent nanocatalysts.

## Methods

**Reagents and chemicals.** Palladium (II) acetylacetonate ($Pd(acac)_2$, 99%), nickel (II) acetylacetonate ($Ni(acac)_2$, 95%), trioctylphosphine (TOP) (90%), Nafion solution (5 wt%), Palladium on activated carbon (Pd/C, 10 wt%) were purchased from Alfa Aesar. Oleylamine (OAm) (>70%) was purchased from Sigma Aldrich. NaOH, ethanol, cyclohexane, toluene, n-hexane and isopropanol were obtained from Beijing Chemical Reagent Company. Ketjen Black was obtained from Shanghai HESEN Electric Company. Milli-Q ultrapure water was utilized through all the experiments.

**Characterization.** TEM images were obtained on a JEM-1200EX (JEOL) transmission electron microscope (TEM) at 100 kV. HRTEM images were recorded via a JEOL JEM-2100F transmission electron microscope operating at 200 kV. Powder XRD patterns were recorded on a Bruker AXS D8-Advanced X-ray diffractometer with Cu Kα radiation ($\lambda = 1.5418$ Å). The tested current and voltage were 40 mA and 40 kV, respectively. A $2\theta$ ranging from 25° to 90° was covered in steps of 0.02° with a count time of 2 s. Elemental composition of the NPs was determined using an inductively coupled plasma mass spectrometer (ICP-MS, Perkin Elmer Elan-6000). The X-ray photoelectron spectrum (XPS) was performed on ESCALAB 250 (Thermo-Fisher Scientific, USA).

**Preparation of precursor NPs.** The details for the preparation of the NPs are shown in Supplementary Table 3. The procedures for preparation of the precursor NPs including $Pd_{38}Ni_{49}P_{13}$, $Pd_{38}Ni_{45}P_{17}$ and $Pd_{40}Ni_{43}P_{17}$ are similar. In a typical preparation of $Pd_{38}Ni_{45}P_{17}$ NPs (Pd:Ni = 5:5 in salt precursors), the reaction was carried out under nitrogen flow and magnetic stirring. About 152.3 mg $Pd(acac)_2$, 135.2 mg $Ni(acac)_2$ and 20 ml OAm were added to a 50-ml three-necked round bottom flask. The mixture was then heated at 80 °C for 10 min to make sure all the reactants were totally dissolved. Then, 1.5 ml TOP was added to the blue transparent solution, thereafter the solution colour changed to green, and the temperature was increased to and kept at 120 °C for 30 min. The solution was then heated to 260 °C within 200 s under sufficient stirring and moderate nitrogen flow. At 230 °C, the solution exhibited a sharp change in colour and turned to black. Then, the temperature was maintained at 260 °C for 1 h. After the solution was

cooled to room temperature naturally, the $Pd_{38}Ni_{45}P_{17}$ NPs were collected by centrifugation at 13,000 r.p.m. and washed three times with an ethanol/cyclohexane mixture. The product was dispersed in 5 ml toluene for later use. Under the same condition, 122 mg $Pd(acac)_2$ and 162.3 mg $Ni(acac)_2$ were used to produce $Pd_{31}Ni_{53}P_{16}$ NPs (Pd:Ni = 4:6 in metal precursors), and 183 mg $Pd(acac)_2$ as well as 108.2 mg $Ni(acac)_2$ were adopted to generate $Pd_{47}Ni_{36}P_{17}$ NPs (Pd:Ni = 6:4 in metal precursors).

**Further phosphorization of the precursor NPs.** The details for the further phosphorization are shown in Supplementary Table 4. Caution: because this procedure involves decomposition of phosphine under high temperature that can liberate phosphorus, this reaction should be considered as highly corrosive and flammable, and therefore should only be carried out by appropriately trained person under strictly air-free conditions. The $Pd_{54}Ni_{30}P_{16}$ (Pd:Ni = 6:4 in metal precursors), $Pd_{40}Ni_{43}P_{17}$ (Pd:Ni = 5:5 in metal precursors) and $Pd_{32}Ni_{50}P_{18}$ NPs (Pd:Ni = 4:6 in metal precursors) (Supplementary Table 4) were obtained respectively by further phosphorization of $Pd_{47}Ni_{36}P_{17}$ (Pd:Ni = 6:4 in metal precursors), $Pd_{38}Ni_{45}P_{17}$ (Pd:Ni = 5:5 in metal precursors) and $Pd_{31}Ni_{53}P_{16}$ (Pd:Ni = 4:6 in metal precursors) precursor NPs (Supplementary Table 3). In a typical reaction, 1 ml (0.2 mmol) as-prepared $Pd_{38}Ni_{45}P_{17}$ precursor NPs toluene dispersion and 1 ml TOP was added to 10 ml OAm at 120 °C. This temperature was maintained for 30 min to remove toluene, water and other low-boiling impurities. Then, the solution was heated to and kept at 290 °C for 1 h under nitrogen flow and moderate stirring. After the solution was cooled to room temperature, the $Pd_{40}Ni_{43}P_{17}$ NPs were collected by centrifugation at 13,000 r.p.m. and washed two times with an ethanol/cyclohexane mixture. The product was dispersed in 5 ml n-hexane, and then 1.0 ml of the obtained dispersion was centrifuged and weighted for further use.

**Preparation of NPs/C catalyst ink.** Seven milligrams of Ketjen Carbon were placed in centrifuge tubes before 3 ml n-hexane was added. The mixture was sonicated for 30 min to ensure the formation of good suspension. Then, 2 ml NPs n-hexane dispersion, containing approximately 7 mg NPs, was added into the Ketjen Carbon suspension. The mixture was then sonicated for 60 min to allow the NPs to be transferred onto the carbon support, which was indicated by the colourless supernatant. About 40 ml of ethanol was added and the mixture was sonicated for further 30 min and centrifuged at 12,000 r.p.m. for 15 min. The colourless supernatant was discarded. The process was repeated twice. The NPs/C product was dried at 60 °C for 12 h, and dispersed in the mixture of ultrapure water, isopropanol and Nafion solution (5 wt%) (v/v/v 3:3:0.2) by sonicated to yield a well-dispersed suspension as catalyst ink with a concentration of $2.4\,mg\,ml^{-1}$. To make the same Pd loading on electrode ($20\,\mu g\,cm^{-2}$) as that of nanocatalysts, the concentration of commercial Pd/C catalyst ink is $4.8\,mg\,ml^{-1}$.

**Electrocatalytic measurements.** CV measurements were carried out in a three-electrode cell using electrochemical workstation (CHI 660E, CH Instrument, Inc.). Our experiments were performed with a saturated calomel electrode (SCE) as the reference electrode. It was calibrated from $E$(RHE, reversible hydrogen electrode) from $E$(SCE) by following the formula $E(RHE) = E(SCE) + 0.254 + 0.05916 \times pH$. The calibration was performed in the high purity hydrogen saturated electrolyte with a Pt foil as the working electrode. As shown in Supplementary Fig. 16, the average of the two potentials where the current crossed zero was taken to be the thermodynamic potential. The drop-casting films of catalysts on glassy carbon electrode (GCE, diameter = 3 mm) served as working electrodes. A special glassy carbon (GC) electrode and saturated calomel electrode (SCE) with a salt bridge were used as the counter and reference electrodes, respectively. Before CV measurements, 3 μl of catalyst ink was dropped onto the polished GCE and evaporated to dry at room temperature. The concentrations of Pd in the catalyst inks were confirmed by inductively coupled plasma mass spectrometer (ICP-MS). All of the CV measurements were obtained at room temperature. The electrolyte solutions were purged with high-purity nitrogen for at least 30 min before use. The working electrode was initially cycled between 0.08 and 1.58 V at $100\,mV\,s^{-1}$ in 1.0 M NaOH for several cycles to remove the residual ligands on catalyst surface. Afterwards, for the EOR measurement, the working electrodes were subject to CV scans between 0.18 and 1.38 V at $100\,mV\,s^{-1}$ in 1.0 M NaOH and 1.0 M ethanol. The chronoamperometry measurements were conducted at 0.8 V in the solution of 1.0 M NaOH and 1.0 M ethanol. For CO-stripping tests, CO oxidation experiments were carried out in the solution of 1.0 M NaOH. Before the test, the solution was purged with nitrogen for 30 min and then was bubbled with CO gas (99.9%) for 15 min at 0.18 V to achieve the maximum coverage of CO at the Pd active centres. The residual CO in the solution was excluded by nitrogen for 30 min.

**Data availability.** Data supporting the findings of this study are available within this article and its Supplementary Information file, and from the corresponding author on reasonable request.

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

## Acknowledgements

This work was supported by the National Natural Science Foundation of China (Grant Nos. 21475007, 21675009 and 21275015). We also thank the support from the 'Public Hatching Platform for Recruited Talents of Beijing University of Chemical Technology'.

## Author contributions

L.Y.W. proposed the research direction and guided the project. L.C., H.L.Z. and Y.G.C. designed and performed the experiments. L.L.L. performed the DFT calculation. Y.H., Y.D.L. and L.Y.W. analysed and discussed the experimental results, and drafted the manuscript. All the authors checked the manuscript.

## Additional information

**How to cite this article**: Chen, L. et al. Improved ethanol electrooxidation performance by shortening Pd–Ni active site distance in Pd–Ni–P nanocatalysts. *Nat. Commun.* **8,** 14136 doi: 10.1038/ncomms14136 (2017).

