## [Peer Review File · Nature Communications]

Reviewers' comments:

Reviewer #1 (Remarks to the Author):

In this manuscript, the authors report a strategy for enhancing ethanol electrooxidation (EOR) catalytic performance by shortening the distance between Pd and Ni active sites of Pd-Ni-P ternary nanocatalysts. Generally, to promote the catalytic performance of EOR, extensive efforts have been devoted to improving the atom usage efficiency of noble metals and/or incorporation of oxophilic metal for facilitating formation of OH radicals. This work presents a viewpoint for enhancing the catalytic performance through surface atom engineering. Considering that free OH radicals and CH₃CO radicals are generated on Ni and Pd active sites, respectively, the author carried out the surface atom engineering and made Ni and Pd atoms close to each other so that to facilitate the rate-determining step of EOR. The Pd-Ni-P nanocatalysts displayed a activity up to 4.95 A/mgPd, about 7 times as high as that of commercial Pd/C. The results implied that the effect of the distance between noble metal and oxophilic metal active sites on the catalytic performance was crucial, though having yet been systematically investigated by other previously reported work. This design idea is very intriguing. Such design principle, I think, offered a new strategy for catalyst design. I suggest the acceptance of this work with minor revision after following points being addressed.

The specific issues that need to be addressed are as follows.

1. All of the catalytic activity testing is done on a per Pd basis. This is reasonable since Pd is active for EOR. However, Ni-P catalysts used as control experiments seem to be ignored. This reviewer thinks the author should verify that Ni-P catalysts are inactive in EOR. Control samples need to be tested catalytically and reported.
2. The authors commented on the excellent stability for EOR catalysis. However, the structural stability or the surface structures of these ternary materials during or after catalysis is not clear. Only TEM measurements in Supplementary Materials were presented. In order to verify the structure stability of these multicomponent nanocatalysts, I advise conducting PXRD measurements to determine whether the nanocatalyst remained intact after catalytic experiments.
3. The authors used a two-step strategy with further phosphorization to prepare Pd₄₀Ni₄₃P₁₇ nanoparticles. Why not using a one-pot procedure? If there was something special in the viewpoint of this method, please mention that.
4. The authors presented the negligible catalytic activity of Pd-Ni-P NPs (Pd/Ni = 0.4%) in Supplementary Fig.8. The CVs in 1.0 M NaOH and 1.0 M C₂H₅OH at 100 mVs⁻¹ (Supplementary Fig. 8c) should be redrawn in order to clearly present their differences or show the closer inspection.
5. In addition to the reference 36 in the manuscript, there are recently sufficiently articles discussing either precious metal nanocatalysts in general or the effect of oxophilic metals in EOR in particular should be added.

Reviewer #2 (Remarks to the Author):

The authors achieved remarkable improvement in activity of the Pd-Ni-P catalyst for ethanol oxidation in alkaline media, and the catalytic peak mass current was almost 7 time as high as that of pure Pd catalyst. They are trying to theoretically explain the phenomenon from the chemical structures by instrumental analysis and DFT modeling. The work is novel either in catalyst's performance or in theoretical understanding by comparison with previous publications. It would be interest for the scientists in the fields of catalysts, materials, and fuel cells. This article should be published in Nature Communication through minor revisions as follows:

1. The stability test is only 2000 seconds, please complete 20 hours chronoamperometric test for one of the best catalyst to verify the durability of it.

2. The DFT and experimental data show only qualitative analysis of "shorten distance" between Pd and Ni. A quantitative analysis should be stronger and increase quality level of this article. If possible, give quantitative analysis data.
3. What is the effect of phosphorus doping? Is it important too for "shorten distance" between P and Pd, or P and Ni?
4. In the experimental section, the reference electrode is saturated calomel electrode (SCE), but in the reported figures and tables, the reference electrode is RHE. How it is calculated or experimentally corrected?
5. The catalyst activity increase is verified by only comparing the mass current, how it is in peak potential, or what is the peak potential differences?

Reviewer #3 (Remarks to the Author):

In this manuscript, the authors present synthesis of small Pd-Ni-P nanoparticles to be used as the catalyst for ethanol electrooxidation reaction (EOR). They proposed that shortening the distance between Pd and Ni would enhance the catalytic activity and durability of the prepared catalyst. The reported performance is among the highest and the proposed mechanisms sound reasonable. Before suggesting the publication of the manuscript in Nature Communications, this referee would like to invite the authors to properly address the following comments:

1. The work suggests that the Pd sites stabilize the CH₃CO radicals and Ni sites stabilize the OH radicals. Does the P element play any role during the EOR or CO stripping process? Or contribute to the stability of the synthesized nanoparticles? Or just involved in the preparation procedure to obtain nanoparticles with desired size and composition?
2. When the Ni sites in the Pd-Ni-P ternary catalyst are expected to facilitate the formation of OH radicals, will the adsorption of other reaction intermediates such as CH₃CO radicals on the Pd sites be influenced by the adjacent Ni and P atoms?
3. Is there any information about the size distribution of synthesized nanoparticles?
4. From the TEM image in Figure 1, the prepared nanoparticles are composed of small clusters of Pd and Ni-P, and from the XPS analysis in Figure 5, Pd(0), NiO/Ni(OH)₂ and P₂O₅/P(0) are the dominate forms of Pd, Ni and P elements in the synthesized nanoparticles. However, in the section of DFT calculations, the authors employed very simple cluster models (Pd₅, Pd₂Ni₃ and Pd₂Ni₂P) to study the OH adsorption and charge distribution. Could these cluster models containing only five atoms accurately represent the real surfaces of the synthesized nanoparticles?
5. In addition to DFT calculations, is there any other direct evidence to support the proposed catalytic scenario?

Replies to the reviewer's questions about manuscript

Reviewers' comments:

Reviewer #1 (Remarks to the Author):

In this manuscript, the authors report a strategy for enhancing ethanol electrooxidation (EOR) catalytic performance by shortening the distance between Pd and Ni active sites of Pd-Ni-P ternary nanocatalysts. Generally, to promote the catalytic performance of EOR, extensive efforts have been devoted to improving the atom usage efficiency of noble metals and/or incorporation of oxophilic metal for facilitating formation of OH radicals. This work presents a viewpoint for enhancing the catalytic performance through surface atom engineering. Considering that free OH radicals and CH₃CO radicals are generated on Ni and Pd active sites, respectively, the author carried out the surface atom engineering and made Ni and Pd atoms close to each other so that to facilitate the rate-determining step of EOR. The Pd-Ni-P nanocatalysts displayed a activity up to 4.95 A/mgPd, about 7 times as high as that of commercial Pd/C. The results implied that the effect of the distance between noble metal and oxophilic metal active sites on the catalytic performance was crucial, though having yet been systematically investigated by other previously reported work. This design idea is very intriguing. Such design principle, I think, offered a new strategy for catalyst design. I suggest the acceptance of this work with minor revision after following points being addressed.

The specific issues that need to be addressed are as follows.

1. All of the catalytic activity testing is done on a per Pd basis. This is reasonable since Pd is active for EOR. However, Ni-P catalysts used as control experiments seem to be ignored. This reviewer thinks the author should verify that Ni-P catalysts are inactive in EOR. Control samples need to be tested catalytically and reported.

Response: Thanks for the kind advice. The Ni-P catalysts were prepared and their activity toward EOR was also evaluated. The results indicated that the Ni₁₂P₅ nanocatalysts characterized by TEM and XRD (shown in Figure S9 a, b), displayed merely the oxidation/reduction peak of Ni element (Figure S9 c) and no ethanol oxidation peak was observed in the presence of ethanol (Figure S9 d), which demonstrated that the Ni-P nanoparticles are inactive for EOR.

Supplementary Figure 9. TEM, XRD and catalytic activity measurements of Ni_{12}P_5 nanocrystals. (a) TEM image and (b) XRD patterns of the as-prepared Ni_{12}P_5 nanocrystals; CVs of Ni_{12}P_5 nanocrystals in (c) 1 M NaOH and (d) 1 M NaOH + 1.0 M $\text{C}_2\text{H}_5\text{OH}$, respectively.

2. The authors commented on the excellent stability for EOR catalysis. However, the structural stability or the surface structures of these ternary materials during or after catalysis is not clear. Only TEM measurements in Supplementary Materials were presented. In order to verify the structure stability of these multicomponent nanocatalysts, I advise conducting PXRD measurements to determine whether the nanocatalyst remained intact after catalytic experiments.

Response: As suggested by the reviewer, the PXRD of the $\text{Pd}_{40}\text{Ni}_{43}\text{P}_{17}$ NPs after catalytic experiments were performed. As indicated in Figure S-1, the XRD pattern of Pd-Ni-P nanocatalysts after electrocatalytic measurement for 2000s were well remained, indicating a good stability of these nanocatalysts.

Figure S-1. XRD patterns of the Pd-Ni-P nanocatalyst before and after catalytic activity measurement

3. The authors used a two-step strategy with further phosphorization to prepare Pd₄₀Ni₄₃P₁₇ nanoparticles. Why not using a one-pot procedure? If there was something special in the viewpoint of this method, please mention that.

Response: Thanks for this good advice. We did attempt to use one-pot method to prepare the Pd-Ni-P ternary nanoparticles under 290 °C. As compared with two-step method, it was found that larger nanoparticles (8 ± 4 nm) with broad size distribution were obtained. Moreover, these nanoparticles are prone to aggregation. Therefore, we used a two-step method to fabricate the Pd-Ni-P ternary NPs.

Figure S-2. TEM images (a) and size distribution (b) of Pd-Ni-P ternary nanoparticles prepared through one-pot strategy.

4. The authors presented the negligible catalytic activity of Pd-Ni-P NPs (Pd/Ni = 0.4%) in Supplementary Fig.8. The CVs in 1.0 M NaOH and 1.0 M C₂H₅OH at 100 mVs⁻¹ (Supplementary Fig. 8c) should be redrawn in order to clearly present their differences or show the closer inspection.

Response: Many thanks. We had revised the Figure according to the suggestion. As shown in Supplementary Figure 12 in the revised manuscript, still negligible catalytic activity toward EOR was observed, implying that highly dispersed Pd with low-dosage hardly showed catalytic activity, which may be attributed to the high sensitivity to CO-poisoning of the nanocrystals.

5. In addition to the reference 30 in the manuscript, there are recently sufficiently articles discussing either precious metal nanocatalysts in general or the effect of oxophilic metals in EOR in particular should be added.

Response: Thanks. The related new article has been added in the revised manuscript, to give a more comprehensive overview in this field.

Reviewer #2 (Remarks to the Author):

The authors achieved remarkable improvement in activity of the Pd-Ni-P catalyst for ethanol oxidation in alkaline media, and the catalytic peak mass current was almost 7 times as high as that of pure Pd catalyst. They are trying to theoretically explain the phenomenon from the chemical structures by instrumental analysis and DFT modeling. The work is novel either in catalyst's performance or in theoretical understanding by comparison with previous publications. It would be interesting for the scientists in the fields of catalysts, materials, and fuel cells. This article should be published in Nature Communication through minor revisions as follows:

1. The stability test is only 2000 seconds, please complete 20 hours chronoamperometric test for one of the best catalysts to verify the durability of it.

Response: The durability of nanocatalysts was evaluated for 20 hours (Supplementary Figure 7) and results suggested that after 20 h test, the Pd₄₀Ni₄₃P₁₇ NPs still show a catalytic activity of 11.23 mA mg⁻¹_{Pd}, 12.83 times higher than that for Pd/C (0.87 mA mg⁻¹_{Pd}).

Supplementary Figure 7. Chronoamperometry curves measured in 1.0 M NaOH + 1.0 M C₂H₅OH (the corresponding potential was held at 0.8 V during the measurements).

2. The DFT and experimental data show only qualitative analysis of “shorten distance” between Pd and Ni. A quantitative analysis should be stronger and increase quality level of this article. If possible, give quantitative analysis data.

Response: It is a very good advice. We were aware of this issue soon after submission of this manuscript and we attempted to give a quantitative analysis ever since. It shall be noted that the exact distance between Ni and Pd is very hard to control during the theoretical calculations considering that the Pd-Ni-P are amorphous catalysts. So, we

utilized a simplified model, namely, Pd-Ni system, for the DFT calculation. To understand the contribution of Pd-Ni distance to the catalytic performance, we further constructed three kinds of nonperiodic slab models and calculated the energy profile of rate-determining step for EOR, i.e. combination reaction between OH radicals (on Ni) and CH₃CO radicals (on Pd) at the PBE/DNP level of theory. Stable absorption geometries of *CH₃CO, *OH and the combination transition states on the catalyst are calculated and shown as below.

Briefly, as shown in the Figure S-3, the energy barrier of the combination reaction on three different catalyst models are calculated to be 32.16, 25.72, 20.49 kcal/mol, respectively. As expected, the energy barrier is significantly decreased with the decrement of distance between Pd and Ni active sites, suggesting that the catalytic performance of nanocatalysts with short Pd-Ni distance is greatly enhanced, which is in good agreement with our experimental results.

Figure S-3. The PBE/DNP calculated energy (kcal/mol) profiles for the combination reaction between OH and CH₃CO radicals on nanocatalysts with different Pd-Ni distance.

3. What is the effect of phosphorus doping? Is it important too for “shorten distance” between P and Pd, or P and Ni?

Response: The phosphorus doping actually has played more than one roles on this system. Firstly, phosphorus doping could efficiently control the size and morphology of the Pd-Ni-P ternary nanoparticles. As indicated in Supplementary Figure 10, in the absence of TOP reagent, only Pd-Ni lump was formed, indicating that the phosphorus doping play crucial roles in the formation of well-dispersed Pd-Ni-P ternary nanoparticles. Secondly, the phosphorus doping process would facilitate formation of phosphides, which then increased the stability of Pd-Ni system, especially under alkali conditions. Thirdly, the phosphorus doping might tune the energy level of Pd (XPS results), as enhanced catalytic activities toward EOR were observed for the Pd-Ni-P ternary nanoparticles.

Supplementary Figure 10. TEM image of Pd-Ni aggregate prepared in the absence of TOP. The synthesis procedures followed the same protocol for preparation of Pd-Ni-P NPs, only without adding TOP reagent.

With respect to the significance of the Pd-P and Ni-P distance, the experimental results (HRTEM images and elemental mapping analysis) indicated that in the Pd-Ni-P system, Ni and P species are homogeneously distributed, suggesting that the phosphorus (P) element mainly combines with Ni rather than Pd and the Pd species are presented as big domains. Thus, it could be considered as Pd-dispersion in Ni-P matrixes. On basis of this observation, the P-Ni distance could be considered to be constant and the modulation of Pd-Ni distance, in some degree, is equivalent to the regulation of P-Pd distance. Therefore, it could be said that the P-Pd distance is important as well.

4. In the experimental section, the reference electrode is saturated calomel electrode (SCE), but in the reported figures and tables, the reference electrode is RHE. How it is calculated or experimentally corrected?

Response: Sorry for the confusion. Our experiments were performed with a saturated calomel electrode (SCE) electrode as the reference electrode. It was calibrated to $E(\text{RHE})$ from $E(\text{SCE})$ by following the formula $E(\text{RHE}) = E(\text{SCE}) + 0.254 + 0.05916 \cdot \text{pH}$. The calibration was performed in the high purity hydrogen saturated electrolyte with a Pt foil as the working electrode. As shown in Supplementary Figure 15, the average of the two potentials where the current crossed zero was taken to be the thermodynamic potential.

Supplementary Figure 15. CVs result of RHE calibration in 0.5 M H₂SO₄ solution

5. The catalyst activity increase is verified by only comparing the mass current, how it is in peak potential, or what is the peak potential differences?

Response: Thanks for the good advice. We totally agree that the mass current and peak potential are all important factors for the performance evaluation of a catalyst. Here, for better comparison, we have listed the values of peak potentials (Figure 2c) in the Table S-1. With respect to the mass current, as discussed in the manuscript, the Pd₄₀Ni₄₃P₁₇ displayed the best activity. In terms of peak potential, the Pd₃₈Ni₄₅P₁₇ and Pd₄₀Ni₄₃P₁₇ have the same peak potential at 0.84 V, indicating that the phosphorization process mainly has contribution to the mass current performance. It is of noted that the peak potential of heterodimers Pd₃₈Ni₄₉P₁₃ is very close to that of Pd/C, implying that these two own similar metallic nature. As for Pd₃₈Ni₄₅P₁₇ and Pd₄₀Ni₄₃P₁₇, the peak potential has a slight positive shift as compared to Pd/C. Similar phenomenon was also observed in the previously reported work (*Angew. Chem.-Int. Edit.* **2015**, *54* (44), 13101; *Angew. Chem.-Int. Edit.* **2014**, *53* (1), 122; *Angew. Chem.-Int. Edit.* **2015**, *54* (12), 3669).

Moreover, as shown in Figure 6d, it is clear that the onset and peak potentials for the electrocatalytic oxidation of CO on both Pd₄₀Ni₄₃P₁₇ and Pd₃₈Ni₄₅P₁₇ NPs are more negative than that on commercial Pd/C (1.0 V), indicating an increased CO oxidation activity and good CO-antipoisoning ability of these NPs. Meanwhile, with the decrement of Pd-Ni distance, the peak potential for CO oxidation became negative, suggesting a better CO-antipoisoning ability.

Figure 2c. c) CVs of Pd-Ni-P NPs with different Pd-Ni distance, and commercial Pd/C in 1.0 M NaOH and 1.0 M C₂H₅OH at 100 mVs⁻¹.

Table S-1. Mass current activity, peak potential and onset potential of all the tested catalysts

Catalyst	Mass current activity (A/mg _{Pd})	Peak potential (V)
Pd/C	0.725	0.8
Pd ₃₈ Ni ₄₉ P ₁₃	4.119	0.79
Pd ₃₈ Ni ₄₅ P ₁₇	4.416	0.84
Pd ₄₀ Ni ₄₃ P ₁₇	4.945	0.84

Reviewer #3 (Remarks to the Author):

In this manuscript, the authors present synthesis of small Pd-Ni-P nanoparticles to be used as the catalyst for ethanol electrooxidation reaction (EOR). They proposed that shortening the distance between Pd and Ni would enhance the catalytic activity and durability of the prepared catalyst. The reported performance is among the highest and the proposed mechanisms sound reasonable. Before suggesting the publication of the manuscript in Nature Communications, this referee would like to invite the authors to properly address the following comments:

1. The work suggests that the Pd sites stabilize the CH₃CO radicals and Ni sites stabilize the OH radicals. Does the P element play any role during the EOR or CO stripping process? Or contribute to the stability of the synthesized nanoparticles? Or just involved in the preparation procedure to obtain nanoparticles with desired size and composition?

Response: Many thanks. As suggested by the reviewer, the P element indeed plays several roles in the Ni-Pd-P system, both in the preparation procedures and catalysis process. As discussed earlier, the P element could efficiently control the size and morphology of the Ni-Pd-P ternary nanoparticles. In the absence of P element, merely Pd-Ni alloy lump was formed (Figure S-4a). In another aspect, the P element did play important roles for EOR in this work. By using different method, we managed to synthesize Pd-Ni NPs, which was confirmed by TEM (Figure S-4b) and XRD (Figure S-4c). Yet, when employing the as-prepared Pd-Ni alloy for EOR, merely low catalytic activity was observed (Figure S-4, d-f), indicating the crucial importance of P element for the excellent catalytic activity of Pd-Ni-P NPs toward EOR.

Figure S-4. (a) TEM image of Pd-Ni aggregates; The synthesis procedures followed the same protocol for preparation of Pd-Ni-P NPs, only without adding TOP reagent; TEM image (b) and XRD pattern (c) of Pd-Ni alloy NPs; (d) CVs obtained on Pd-Ni in 1.0 M NaOH solution at a scan rate of 100 mVs⁻¹; (e) CVs of Pd-Ni in 1.0 M NaOH and 1.0 M C₂H₅OH at 100 mVs⁻¹; (f) Chronoamperometry curves measured in 1.0 M NaOH and 1.0 M C₂H₅OH (the corresponding potential was held at 0.8 V during the measurements).

The detailed procedures of preparation of Pd-Ni alloy are as follows: Pd(acac)₂ and Ni(acac)₂ (Pd/Ni = 1:1) were dissolved in the mixture solvent of 10 mL OAm and 200 μL ODE. 3 mmol Ammonia borane was then injected under 60 °C as reduction and nucleation reagent. Then the mixture was heated up to 270 °C and maintained for 1 hour. Then the Pd-Ni alloys were obtained by centrifugation.

2. When the Ni sites in the Pd-Ni-P ternary catalyst are expected to facilitate the formation of OH radicals, will the adsorption of other reaction intermediates such as CH₃CO radicals on the Pd sites be influenced by the adjacent Ni and P atoms?

Response: In theory, the adsorption of CH₃CO radicals on the Pd sites would be influenced by the adjacent Ni and P atoms since the possible steric effect may arise from the adjacent atoms. However, so far, we have not had direct evidence for this assumption. Herein, in the Pd-Ni-P ternary nanocatalysts, the Pd exists as clusters instead of single atoms. So, the steric effect is not so obvious. When we decrease the Pd dosage to 0.4% to decrease the size of Pd clusters, the EOR activity was very weak, which may be attributed to the steric hindrance and easy CO-poisoning of Pd in so small size.

3. Is there any information about the size distribution of synthesized nanoparticles?

Response: As suggest by reviewer, we had evaluated the size distribution of the as-synthesized nanoparticles by measuring the size of nanoparticles and the statistics analysis were shown in Supplementary Figure 1 and Figure S-5. As shown in Supplementary Figure 1, the synthesized nanoparticles (Pd₃₈Ni₄₉P₁₃, Pd₃₈Ni₄₅P₁₇ and Pd₄₀Ni₄₃P₁₇) have similar sizes with narrow distribution around 5 nm.

Supplementary Figure 1. Size distribution of these Pd-Ni-P ternary NPs. The size distribution profiles of Pd₃₈Ni₄₉P₁₃ (a), Pd₃₈Ni₄₅P₁₇ (b) and Pd₄₀Ni₄₃P₁₇ (c), respectively. The average sizes of these NPs are 5.5 ± 1.0 nm, 5.3 ± 1.0 nm, 5.3 ± 0.5 nm, respectively.

With respect to the nanoparticles having different Pd/Ni feeding ratios, the sizes have a slight differences, namely 4.5±0.8 nm, 5.3±1.0 nm, 6.3±1.5 nm for Pd₄₇Ni₃₆P₁₇ (Pd/Ni = 6:4), Pd₃₈Ni₄₅P₁₇ (Pd/Ni = 5:5) and Pd₃₁Ni₅₃P₁₆ (Pd/Ni = 4:6). After phosphorization, their sizes changed into 3.9±0.5 nm, 5.3±0.5 nm, 5.6±1.0 nm, respectively, which indicated that the phosphorization procedure made the size distributed more uniformly.

Figure S-5. The size distribution profile of Pd₄₇Ni₃₆P₁₇ (Pd/Ni = 6:4), Pd₃₈Ni₄₅P₁₇ (Pd/Ni = 5:5) and Pd₃₁Ni₅₃P₁₆ (Pd/Ni = 4:6) before (a-c) and after (d-f) phosphorization.

4. From the TEM image in Figure 1, the prepared nanoparticles are composed of small clusters of Pd and Ni-P, and from the XPS analysis in Figure 5, Pd(0), NiO/Ni(OH)₂ and P₂O₅/P(0) are the dominate forms of Pd, Ni and P elements

in the synthesized nanoparticles. However, in the section of DFT calculations, the authors employed very simple cluster models (Pd₅, Pd₂Ni₃ and Pd₂Ni₂P) to study the OH adsorption and charge distribution. Could these cluster models containing only five atoms accurately represent the real surfaces of the synthesized nanoparticles?

Response: Indeed, we had applied a simplified model in DFT calculation, which can hardly represent the real surfaces of the synthesized nanoparticles. Suggested by our experimental results, the degree of crystallinity decreased with the increase of phosphorization time. The topic catalyst (Pd₄₀Ni₄₃P₁₇) with the best electrocatalytic performance has the amorphous nature, which was confirmed by X-ray diffraction (XRD) analysis and in well agreement with previous experimental report (ref. 28 in the manuscript). With respect to theoretical calculation of these amorphous catalysts, the structure characterization is a challenge, since the routine catalyst models such as slab model derived from crystalline structure are unsuitable anymore. Therefore, we had to use a model that is close to the real state. The cluster models, to the best of our knowledge, are the most appropriate alternative to perform the DFT calculations so as to elucidate the OH⁻ adsorption and OH radical production as well as explain the different catalytic activities of the investigated ternary catalysts.

To obtain the critical knowledge about the site preference of OH⁻ adsorption over Pd, Ni, P atoms, the formation and dissociation of *OH, we perform the DFT calculations with the relatively accurate hybrid density functional (B3LYP). To get a compromise between the calculation accuracy and time cost, the simple cluster model (Pd₂Ni₂P) was adopted because of the similar atomic ratio to the as-prepared Pd-Ni-P catalyst (ICP-MS results) in this work. Pd₅ and Pd₂Ni₃ cluster models were also comparatively studied to explore the effects of P, Ni elements on the catalytic mechanism. The theoretical results based on these cluster models indicate the OH⁻ shows preferential adsorption on Ni atoms, this is in good agreement with experimental result presented in ref. 26 in the manuscript. In addition, the charge analysis based on the cluster model demonstrated the consistent charge distributions with our experimental XPS observation in this work. On the basis of this result, we think that these cluster models adopted in this work can give a reasonable and relatively accurate description of the investigated catalysts, catalytic performance and mechanism.

As mentioned above, we also constructed the Pd-Ni system model for the DFT calculation. As shown in the Figure S-3, the energy barrier of the combination reaction on three different catalyst models are calculated to be 32.16, 25.72, 20.49 kcal/mol, respectively. As expected, the energy barrier is significantly decreased with the decrement of distance between Pd and Ni active sites, suggesting that the catalytic

performance of nanocatalysts with short Pd-Ni distance is greatly enhanced, which is in good agreement with our experimental results.

5. In addition to DFT calculations, is there any other direct evidence to support the proposed catalytic scenario?

Response: Many thanks. We do hope find more direct experiment data to support this hypothesis, but it is not available yet. In the current work, as shown in the HRTEM images of the as-prepared Pd-Ni-P nanocatalysts, the Pd domains became smaller and smaller, and the distance between Pd and Ni was decreased by prolonging the phosphorization time. Meanwhile, the EOR catalytic activity was promoted step by step. These results were in agreement with the DFT calculation. In addition, with the decrement of Pd-Ni distance, the CO tolerance ability was also enhanced (Figure 6d).

REVIEWERS' COMMENTS:

Reviewer #1 (Remarks to the Author):

I have checked this revised manuscript carefully as well as the authors response to other reviewers comments. I think the quality of this revised version is much better than the original one. In my own opinion, this manuscript can be accepted now.

Reviewer #2 (Remarks to the Author):

According to the reviewers' comments, the authors have careful addressed, and done some necessary new researches and experiments. All answers are satisfactory. The revised manuscript apparently has been improved in quality, which meets the requirements by Nature Communication. I agree this manuscript to be published in Nature Communication with the present status or some minor revisions.

Reviewer #3 (Remarks to the Author):

In this manuscript, the authors propose that the catalytic activity of Pd-Ni-P ternary nanoparticles towards the ethanol electrooxidation reaction (EOR) could be significantly enhanced by shortening the distance between Pd and Ni sites. In the revised manuscript, the authors addressed previously raised questions and supplemented persuasive experiments and calculations to further support their arguments, therefore I recommend this revised manuscript to be accepted by Nature Communications.

Replies to the reviewer's questions about manuscript

Reviewers' comments:

Reviewer #1 (Remarks to the Author):

I have checked this revised manuscript carefully as well as the authors response to other reviewers comments. I think the quality of this revised version is much better than the original one. In my own opinion, this manuscript can be accepted now.

Response: Many thanks for the kind comment.

Reviewer #2 (Remarks to the Author):

According to the reviewers' comments, the authors have careful addressed, and done some necessary new researches and experiments. All answers are satisfactory. The revised manuscript apparently has been improved in quality, which meets the requirements by Nature Communication. I agree this manuscript to be published in Nature Communication with the present status or some minor revisions.

Response: We are delight to know that you were satisfied with the revisions. Many thanks.

Reviewer #3 (Remarks to the Author):

In this manuscript, the authors propose that the catalytic activity of Pd-Ni-P ternary nanoparticles towards the ethanol electrooxidation reaction (EOR) could be significantly enhanced by shortening the distance between Pd and Ni sites. In the revised manuscript, the authors addressed previously raised questions and supplemented persuasive experiments and calculations to further support their arguments, therefore I recommend this revised manuscript to be accepted by Nature Communications.

Response: Thanks for the comment.